# FREQUENCY DECOMPOSITION IN NEURAL PROCESSES

## ABSTRACT

Neural Processes are a powerful tool for learning representations of function spaces purely from examples, in a way that allows them to perform predictions at test time conditioned on so-called context observations. The learned representations are finite-dimensional, while function spaces are infinite-dimensional, and so far it has been unclear how these representations are learned and what kinds of functions can be represented. We show that deterministic Neural Processes implicitly perform a decomposition of the training signals into different frequency components, similar to a Fourier transform. In this context, we derive a theoretical upper bound on the maximum frequency Neural Processes can reproduce, depending on their representation size. This bound is confirmed empirically. Finally, we show that Neural Processes can be trained to only represent a subset of possible frequencies and suppress others, which makes them programmable band-pass or band-stop filters.

## 1 INTRODUCTION

Neural Processes (Garnelo et al., 2018a;b) are a class of models that can learn a distribution over functions, or more generally a function space. In contrast to many other approaches that do the same, for example Bayesian Neural Networks, Neural Processes learn an *explicit representation* of such a function space, which allows them to condition their predictions on an arbitrary number of observations that are only available at test time. This representation is finite-dimensional, while function spaces are infinite-dimensional, and so far it has not been understood how they are able to bridge this gap and under what conditions they can successfully do so.

Our work reveals how Neural Processes learn to represent infinite-dimensional function spaces in a finite-dimensional space, and in the process describes constraints and conditions that decide what kinds of function spaces can be represented. We begin with an observation that prior art in the context of learning on sets can be reinterpreted from a signal-processing perspective, which allows us to derive a theoretical upper bound on the *frequencies*, i.e. Fourier components, of functions that can be represented. We subsequently confirm this bound empirically, which suggests that the learned representations should contain a notion of frequency. To further investigate this hypothesis, we continue with a visualization of the learned representations, which reveals that Neural Processes can decompose a function space into different frequency components, essentially finding a representation in Fourier space without any explicit supervision on the representations to elicit such behaviour. As further evidence of this we train Neural Processes to represent only certain frequencies, which results in them suppressing those frequencies that were not observed in the training data. Our contributions can be summarized as follows[1]:

- We derive a theoretical upper bound on the signal frequency Neural Processes of a given representation size can reconstruct. As we show, the bound is observed either in the expected way—by suppressing high frequencies—or by implicitly limiting the signal interval.

- We investigate learned representations qualitatively, presenting evidence that Neural Processes perform a frequency decomposition of the function space, akin to a Fourier transform. This behaviour is not incentivized externally but rather emerges *naturally*.

---

[1]The complete source code to reproduce our experiments is available at `https://github.com/***`

- We show that by choosing the training distribution appropriately, Neural Processes can be made to represent certain frequencies and suppress others, which turns them into *programmable band-pass or band-stop filters*.

## 2 BACKGROUND

Neural Processes (Garnelo et al., 2018a;b) are maps $P : C, X \to Y$, where $C$ is a set of tuples $\{(x, f(x))\}_{c=1}^{N} =: (\mathbf{x_c}, \mathbf{f}(\mathbf{x_c}))^2$ with arbitrary but positive cardinality $N$, and $f \in \mathcal{F} : X \to Y$. $C$ is often called the *context*, because Neural Processes perform predictions for values $x_t \in X$ ($t$ for *target*), conditioned on these points. $\mathcal{F}$ is the function space we would like to find a representation of. Note that some sources define function spaces as any set of functions with a shared domain and co-domain, while others require them to be vector spaces as well. We don't concern ourselves with this distinction and further restrict our work to $X = Y = \mathbb{R}$, because it allows us to visualize learned representations. We only look at the original Neural Processes, namely the deterministic Conditional Neural Processes (CNP) (Garnelo et al., 2018a) and the variational Neural Processes (NP) (Garnelo et al., 2018b), because newer contributions in the field work in ways that preclude them from being analyzed in the same way. We discuss this further in Section 5. In CNPs and NPs, the map $P$ is separated into two parts, a so called *encoding* $E : C \to Z$ and a *decoding* or *generating* part $G : Z, X \to Y$. $Z$ is referred to as the *representation* or *latent space*. To allow Neural Processes to approximate arbitrary[3] function spaces $\mathcal{F}$, $E$ and $G$ are typically chosen to be powerful approximators, specifically neural networks, as the name suggests.

The defining characteristic of CNPs and NPs is that $E$ encodes individual pairs $(x, f(x))$ from the context *separately*, and the resulting representations are averaged to form a *global representation*, meaning one that is independent of the target points $\mathbf{x_t}$ at which we then evaluate the Neural Process. This is often not the case in later work, for example in Attentive Neural Processes (Kim et al., 2019), where the individual representations are instead aggregated using an attention mechanism that depends on $\mathbf{x_t}$. In CNPs the representations are deterministic, while in NPs they parametrize mean and (log-)variance of a Gaussian distribution, so the latter are trained using variational inference. For details on implementation and training we refer to Appendix A.1. Our work will investigate how these global representations, which are *finite-dimensional*, represent *infinite-dimensional* function spaces.

As stated above, $E$ and by extension the Neural Process $P$ acts on *set-valued* inputs. This is contrary to the vast majority of machine learning work where inputs are vectors of fixed dimension and ordering. Recall that sets are permutation invariant, so we must ensure that the same is true for the output of $E$. It is easy to see that this is given when we average individual encodings, but Zaheer et al. (2017) show that it is in fact the only way to ensure it: $E$ is permutation-invariant if and only if it has a so-called *sum-decomposition*, i.e. it can be represented in the form

$$E(\mathbf{x}) = \rho \left( \sum_{i=1}^{N} \phi(x_i) \right) \tag{1}$$

where $\rho, \phi$ are appropriately chosen functions. Wagstaff et al. (2019) further show that to be able to represent all continuous permutation-invariant functions on sets with a cardinality of at most $N$, the dimension of the image $Z$ must at least be $N$. This will become relevant in the following section.

## 3 AN UPPER BOUND ON SIGNAL FREQUENCIES

We mentioned in the previous section that the encoder $E$ in a Neural Process should have a sum-decomposition, so that the global representations are permutation-invariant, as shown in Zaheer et al. (2017). Expanding on this, Wagstaff et al. (2019) show that we require a representation size of at least $N$ to be able to represent arbitrary continuous functions on sets of cardinality smaller or equal to $N$. What these works do not consider are the implications for situations where the elements of

---

[2]We use boldface as a shorthand for *sets*, not vectors.

[3]This will depend on the implementation of $E$ and $G$, and for neural networks $\mathcal{F}$ is practically restricted to continuous and differentiable functions.

the sets are input-output tuples of some function $f$, as it is typically the case in Neural Processes. We will use these previous findings to derive an upper bound on the frequencies $\nu$ any $f \in \mathcal{F}$ may contain so that they can be represented in a Neural Process. In order to do this, we must first define what it means to successfully learn a representation of a function space.

**Definition 3.1** (Representation of Function Spaces in Neural Processes). We say that a Neural Processes $P$ has learned a representation of a function space $\mathcal{F}$, defined on an interval $[a, b] \subset \mathbb{R}$, if, for some error tolerance $\epsilon$, it holds for all $x \in [a, b]$ and for all $f \in \mathcal{F}$, represented as a suitable set of discrete measurements $(\mathbf{x_f}, f(\mathbf{x_f}))$, that $|P((\mathbf{x_f}, f(\mathbf{x_f})), x) - f(x)| < \epsilon$.

That means the learned representation must be such that we can encode a particular element of the function space $f$ into it and are able to reconstruct it up to a predefined error tolerance. The choice of this tolerance is essentially arbitrary, but should reflect that for $g \notin \mathcal{F}$ the reconstructions should generally not be accurate within $\epsilon$. We also write that $f$ is represented as a *suitable* set of discrete measurements, by which we mean that it must be possible to reconstruct $f$ from those measurements.

Switching to signal-processing terminology, we know that to represent a continuous signal as a set of discrete measurements, we need to sample it at points with a distance of at most $\tau = 1/(2\nu_{\max})$, where $\nu_{\max}$ is the maximum frequency component of the signal. This is most commonly known as the Nyquist-Shannon sampling theorem (Whittaker, 1915; Kotelnikov, 1933; Shannon, 1949). For any finite real interval $[a, b]$, this translates to a number of sampling points $N > 2|b - a|\nu_{\max}$. The latter allows us to make a connection to the findings by Wagstaff et al. (2019), so that we can deduce an upper bound on the maximum signal frequency Neural Processes with a given representation size can reconstruct.

**Theorem 3.1** (Maximum Frequency in Neural Process Representations). A Neural Process $P$ with latent dimension $D_r$ can only learn a representation of some function space $\mathcal{F}$ defined on a finite interval $[a, b] \subset \mathbb{R}$ if for all $f \in \mathcal{F}$ with a maximum frequency content $\nu_{\max,\mathrm{f}}$ it holds that:

$$\nu_{\max,\mathrm{f}} < \frac{D_r}{2|b - a|} \tag{2}$$

Note that this means we should in theory be able to represent *any* function space that obeys Eq. (2) to within *arbitrarily small* $\epsilon$. In practice, we will typically have less control over $\mathcal{F}$, and we only find approximate representations. Part of our experiments will test how Neural Processes behave if the signals contain frequencies larger than those allowed by Eq. (2). It should also be noted that the Nyquist-Shannon theorem used for the above derivation assumes equidistant sampling points. During training, we work with *randomly* sampled inputs, but at test time equidistant points are used, as we outline in Appendix A.2.

## 4 EXPERIMENTS & RESULTS

### 4.1 VALIDATION OF THE FREQUENCY BOUND

Our experiments are grouped into three parts. The first experiment seeks to test the validity of the bound we just derived in Eq. (2). In particular, we train Neural Processes with varying representation sizes on two exemplary function spaces, so that for some models the representation size is insufficient to represent all frequencies. The function spaces we base our experiments on are those defined by Gaussian Process priors (for an introduction see for example Rasmussen & Williams (2006)) with an exponentiated-quadratic (EQ) kernel with lengthscale parameter $l$, as well as those defined by random real-valued Fourier series—for details we refer to Appendix A.2. While the Gaussian Process samples have an average Fourier magnitude that smoothly decays to zero, the distribution of Fourier magnitudes is uniform for the Fourier series, as shown in Fig. A.1. The Fourier series space also grants us precise control over the frequency domain, which will be useful in subsequent experiments.

Figure 1 shows example reconstructions in a deterministic Neural Process (CNP) for samples from a Gaussian Process prior with EQ kernel ($l = 0.05$) and from a random Fourier series. For the GP example, the CNP essentially acts like a low-pass filter when the representation size is insufficient, which qualitatively confirms the bound we derived in Eq. (2). Interestingly, the bound can also be observed in a different way: for the Fourier series example, the CNP hardly suppresses high

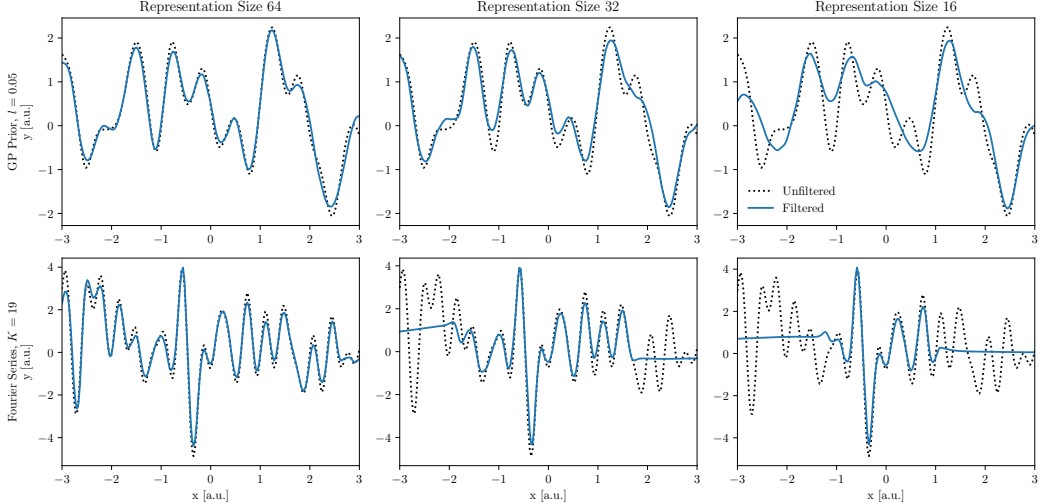

Figure 1: Example reconstructions from a deterministic Neural Process (CNP) with varying representation sizes. For the Gaussian Process example, reducing the representation size removes higher frequency components. For the Fourier series example, the CNP instead limits the effective interval. Both behaviours respect the bound derived in Eq. (2).

frequencies, but instead limits the *effective interval* of the signal, simply ignoring the outer regions of it. Both behaviours are in agreement with the bound in Eq. (2). The Fourier example also serves as a good sanity check: with $K = 19$ (the maximum *angular* frequency) the data has a maximum frequency of $\nu_{\max} = K/(2\pi) = 3.02$. For $D_r = 32$ this would limit the size of the interval to $|b - a| < 5.29$, for $D_r = 16$ to $|b - a| < 2.65$. The reconstructed signal regions in Fig. 1 are a bit narrower, and thus in good agreement with the bound we derived. For a variational Neural Process, we observe the same behaviour, but with stronger dampening of high frequencies in both cases, as seen in Fig. A.2. In Fig. A.3 we show the average reconstruction error for CNPs and NPs of different representation sizes, applied to GP examples with varying lengthscale, which results in a smooth decrease in error for larger representations and larger lengthscale parameters, as one would expect.

## 4.2 How do Neural Processes Represent Function Spaces?

Having found that Neural Processes do indeed observe the bound we derived in Eq. (2), we seek to understand how this happens. To this end, we visualize the learned representations in Neural Processes, which is possible because we restrict ourselves to $X = Y = \mathbb{R}$. Again looking at the two function spaces from the previous experiment, we sample pairs $(x, y)$ on a regular grid ($50 \times 50$) with $x \in [-3, 3]$, which is our training input range, and also $y \in [-3, 3]$ as it suitably covers the value range of outputs. We then encode each pair individually to a representation, thus constructing a map $r_i(x, y)$ for each representation channel. The latter allows us to uncover potential patterns and to gain a better understanding of how Neural Processes learn representations of function spaces.

Figure 2 presents example representation channels for CNPs and NPs, trained on samples from a Gaussian Process with an EQ-kernel ($l = 0.2$) and on random Fourier series. The individual channels were selected to illustrate the general patterns of behaviour we observed. First, we find that representations are almost always anti-symmetric across $y = 0$. This is not surprising, as the function spaces we look at are on average symmetric—in the sense that $f$ and $-f$ will occur with the same probability—so the Neural Process learns the same representation, just with a different sign. More importantly, we find that both NPs and CNPs implicitly form a representation of the input space (i.e. the relevant interval of the function space domain), in the sense that different regions of the input space map to different representation channels. In CNPs this results in an oscillating pattern, with different channels exhibiting different frequencies. In other words, the CNP performs a *frequency decomposition* of the function space, not unlike a Fourier transform. At the

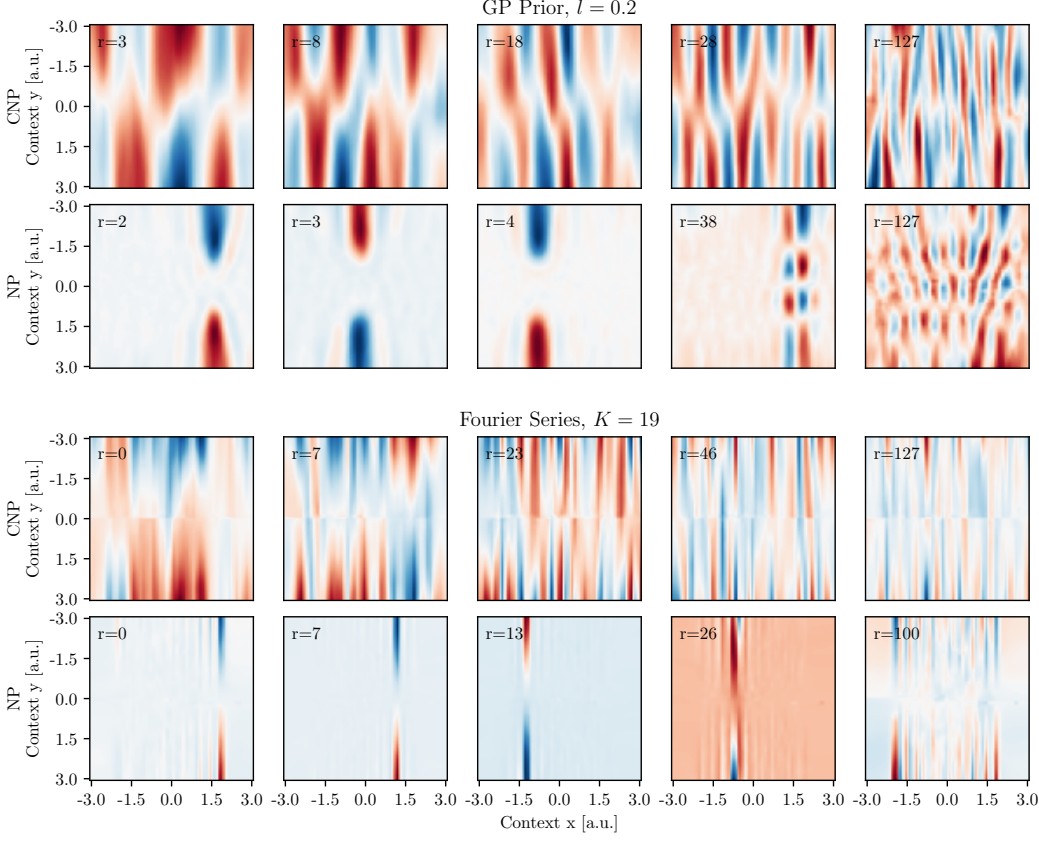

Figure 2: Influence of the context on the learned representations in a deterministic Neural Process (CNP) and a variational Neural Process (NP) with $D_r = 128$ and data coming from a Gaussian Process prior with an EQ kernel with $l = 0.2$ (top half, kernel description in Eq. (10)) or a Fourier series with $K = 19$ as described by Eq. (11) (bottom half). $x$ refers to the input space (i.e. the domain), $y$ to the output space (i.e. the co-domain). Representation channels are ordered by their average magnitude. We find that the learned representations are almost always anti-symmetric across $y = 0$. The CNP exhibits an oscillating behaviour along the x-axis, with varying frequency across representation channels. In contrast, the NP seems to mostly partition the input space into separate regions, only exhibiting oscillating behaviour in channels with lower magnitude. The representation channels in this example were selected for illustrative purposes. Note that each panel is normalized separately, so color values are not comparable.

same time, there is nothing that would enforce orthogonality between the different representation dimensions, and the Fourier series example highlights that we can generally expect a mixture of multiple frequencies for a given dimension. It should be noted that this frequency decomposition emerges *naturally* and is not incentivized externally (e.g. by a special loss).

Even though NPs behaved very similarly to CNPs in the previous section, their learned representations look vastly different from those in a CNP. Instead of a frequency decomposition, they seem to *partition* the input space, so that a given representation dimension is written to by a specific, narrow region of the input space. Only for channels with a low average magnitude (i.e. a large index in Fig. 2) do we find behaviour similar to CNPs. We conclude that NPs can in principle learn a frequency decomposition, but their variational formulation—the only difference to CNPs—disincentivizes it. We show more representations for CNPs and NPs trained on GP data in Fig. A.4 and Fig. A.5, and for CNPs and NPs trained on Fourier series data in Fig. A.6 and Fig. A.7, sorting channels by their average magnitude.

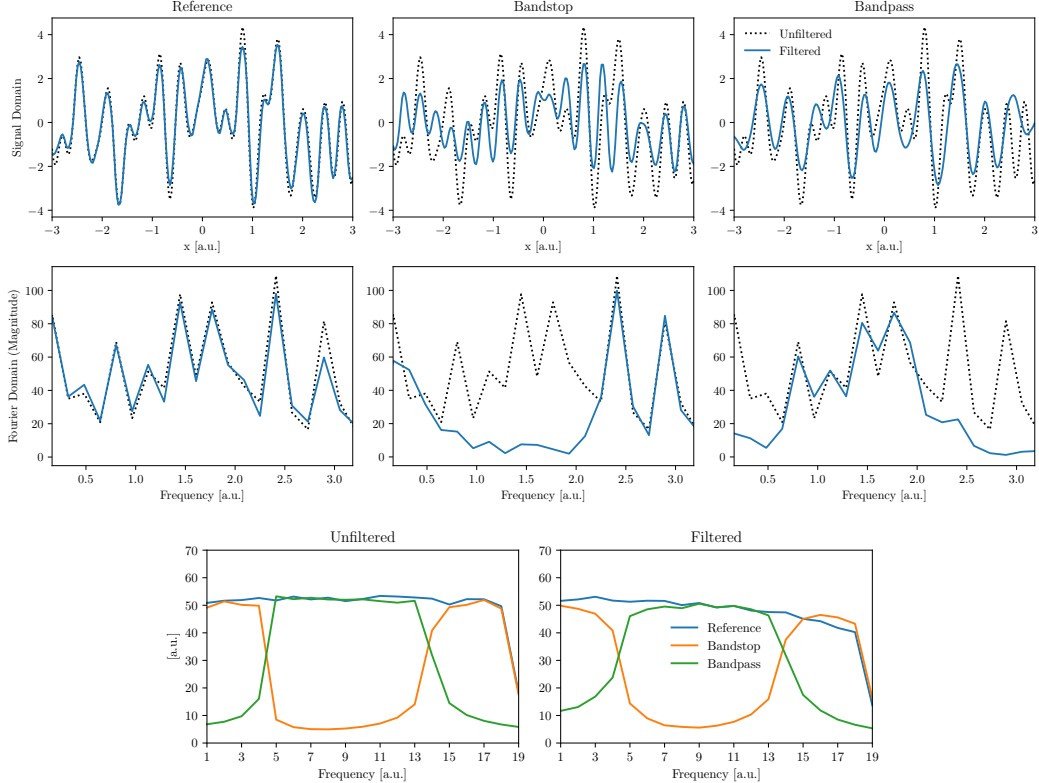

Figure 3: Deterministic Neural Process (CNP) trained to be a band-pass and a band-stop filter. We train three CNP models on Fourier series, where we only show certain frequencies during training for the models that should serve as frequency filters. The bottom row shows the distribution of Fourier components in the training data (left) and after applying each model to the *reference* data (right). The models only learn to represent frequencies seen during training, and as a result act like band-pass and band-stop filters. The top rows show an example of this. See Fig. A.8 for the same experiment using a variational Neural Process.

## 4.3 NEURAL PROCESSES AS BAND FILTERS

Our final experiment is designed to show that we can exert more control over the learned representations, and it will serve as additional evidence that deterministic Neural Processes (CNP) perform a frequency decomposition of the function space they represent. At the same time, it suggests a possible practical application of Neural Processes. We saw in Section 4.1 that CNPs sometimes act like low-pass filters, which could be a useful application, but the emergence of that behaviour is not reliable. We now train CNPs with a sufficiently large representation size ($D_r = 128$) to be band-pass and band-stop filters. To this end, we train the models on the Fourier series defined by Eq. (11), but for the band-stop we set all components $a_k$ to zero for which $5 \leq k \leq 14$, and likewise set all $a_k$ to zero *outside* of that range for the band-pass. We then look at the reconstructions of examples from the original series with all components present. For more details on the training procedure and how we sample points for function evaluation, please see Appendix A.1 and Appendix A.2.

The average Fourier magnitude of the training functions for the different models is given by the bottom left panel in Fig. 3. In the first model (Reference), all components are allowed; in the second (Band-stop), components in the middle of that range are suppressed; in the third (Band-pass) only components in the middle of the range are allowed. We then apply these models to examples from the *reference* data distribution, the result of which can be seen in the bottom-right panel of Fig. 3. The models that are only shown certain frequencies during training will suppress those frequencies that were not found in the training data, meaning they effectively become programmable band-stop or band-pass filters. This is confirmed by the example in the top rows of the figure, where we show

both the signal and its Fourier transform magnitude. Note that one needs to adjust the value range of the reference data before passing them through the band filters to prevent gain in the non-suppressed frequency regions. We give more details in Appendix A.2.

Unfortunately, we were only partly able to elicit the same behaviour in variational NPs. While the trained band-stop filter worked exactly like the CNP band-stop, we were not able to train a band-pass filter. The models collapsed during training, meaning the loss plateaued and no meaningful representations were learned. There is no obvious reason why a band-pass shouldn't work when a band-stop does, so we suspect our hyperparameter configuration was not ideal and that with more tuning it would be possible to train a band-pass as well. The NP results are shown in Fig. A.8.

## 5 RELATED WORK

Neural Processes broadly relate to the topic of *learning distributions of functions*, even though we speak of the less restrictive term *function space* in our work. In this context, *Bayesian Neural Networks* (see for example Neal (1996); Graves (2011); Hernández-Lobato & Adams (2015)) are a popular choice, which place distributions on the weights of a network. However, in doing so they only implicitly represent distributions over functions, while Neural Processes learn an explicit finite-dimensional representation that can be leveraged for predictions, so as to condition on context observations given at test time. Perhaps the most well known class of methods that do the same are *Gaussian Processes* (for an introduction see Rasmussen & Williams (2006)). These are stochastic processes represented by a joint Gaussian distribution over context and target points, defined via the covariance matrix by a kernel. All flexibility of Gaussian Processes to represent different distributions of functions is decided by this kernel, so many works try to learn it (Yang et al., 2015; Wilson et al., 2016b;a; Tossou et al., 2019; Calandra et al., 2016). Even though Neural Processes were originally motivated by Gaussian Processes, they can be understood as orthogonal methods: Gaussian Processes represent a function space using a (potentially learned) kernel, while Neural Processes represent them in a learned finite-dimensional space.

Neural Processes can also be interpreted from the perspective of deep learning on sets, the earliest work in the field being Zaheer et al. (2017). More theoretical contributions were made by Wagstaff et al. (2019), whose work we use to underpin our finding that the representation size in Neural Processes limits the maximum frequency of signals that can be represented. More applied work in the set-learning context has mostly been performed on point-cloud data (Qi et al., 2017b;a; Wu et al., 2019), which can be interpreted as a higher-dimensional instance of learning function spaces. Validating our findings in higher-dimensional spaces is an important direction for future work.

Neural Processes have inspired a number of follow-up works. Perhaps the most well known addition are Attentive Neural Processes (Kim et al., 2019), which replace the averaging of individual representations with a learned attention mechanism (Vaswani et al., 2017). The aggregate representations are thus no longer independent of the target inputs, and no *global representation* is learned. This holds true for most follow-up work. Convolutional Conditional Neural Processes (Gordon et al., 2020) propose to no longer learn a finite-dimensional representation at all and instead work in function space by applying a CNN on suitable and variable discretizations of a kernel density estimate. Similar to ANP, Louizos et al. (2019) propose to not merge observations into a global latent space, but instead learn conditional relationships between them. Singh et al. (2019) and Willi et al. (2019) address the problem of overlapping and changing dynamics in time series data. Relating this to our work, it would be possible to test how the original Neural Processes would represent functions where the average frequency content is not constant over the domain. We leave this investigation for future work. Neural Processes have also been extended to scenarios where the function space maps to entire images, in the form of Generative Query Networks (GQN) (Eslami et al., 2018; Kumar et al., 2018). Employing vastly more powerful decoders, they can (re-)construct unseen views in 3D scenes, which relates Neural Processes to the field of 3D scene understanding, an area that has received a lot of attention more recently (Sitzmann et al., 2019; Engelcke et al., 2020; Mildenhall et al., 2020). Sitzmann et al. (2020) show that periodic activation functions make it easier for networks to learn so-called *implicit representations*—mappings from coordinates to a density, RGB values, etc.. We did in fact try periodic activation functions in our experiments, but found no difference to using $\tanh$-activations. In the same context, Tancik et al. (2020) show that coordinates in Fourier space are often superior to coordinates in signal space to produce fine detail. We interpret this as an indication

that a representation in frequency space is more efficient for many signals, which could explain why Neural Processes implicitly perform a frequency decomposition. Note that the above introduces Fourier features explicitly as a form of inductive bias, while Neural Processes automatically learn this form of representation.

It is well known that neural networks, specifically a MLP with at least one hidden layer, can learn the Fourier transform of an input signal (Gallant & White, 1988). In fact, there have been a multitude of works that exploit this ability in one way or the other, leading to the term *Fourier Neural Networks*. We refer to the recent review by Zhumekenov et al. (2019) for a comprehensive overview. The difference to Neural Processes is that these works typically apply networks directly to a *sequence* of points, while NPs learn a mapping that is only applied to individual (x,y) pairs, the representations of which are averaged. We emphasize again that the frequency decomposition occurs *naturally* in NPs, while these works usually employ direct supervision.

## 6 DISCUSSION

The goal of this work was to gain a better understanding of the mechanisms that allow Neural Processes to form finite-dimensional representations of infinite-dimensional function spaces. To the best of our knowledge, ours is the first work to investigate this question, and our findings are both surprising and meaningful in this context. We first derived a theoretical upper bound on the frequency of signals that can be represented in Neural Processes with a given representation size. We empirically confirmed that the representation size does indeed pose such a limit and that this can result in Neural Processes acting like low-pass filters. Alternatively, models ignore parts of the signal to keep higher frequencies. Both behaviours are in agreement with the derived bound. We then visualized learned representations to understand how the models incorporate the concept of frequency into them. In all cases the models formed an implicit representation of the input space, in the sense that different x-values are mapped to different representation channels. For CNPs, an oscillating pattern emerges, such that different representation channels correspond to different frequencies, from which we concluded that CNPs perform a *frequency decomposition* of the function space they learn to represent. It should be noted that this behaviour emerges *naturally* and is not explicitly encouraged (e.g. by a special loss). In contrast to this, NPs tend to partition the space into more or less disjunct regions. They are still able to learn a frequency decomposition like CNPs, but we assume that the variational training objective makes it harder to do so, as sampling from the representation during training can also be understood as a random perturbation. For VAEs, which are conceptually similar to NPs, it was also suggested that models partition their latent space in way that maximally spreads representations of individual data points under the prior distribution (Rezende & Viola, 2018). Finally, to further test the models' ability to distinguish frequencies and also as an example of possible practical benefits of our findings, we trained CNPs to be band-pass and band-stop filters. This worked extremely well, the Fourier component magnitudes of the training data are essentially "baked" into the models, and any frequency not found therein is subsequently suppressed in reconstructions from the models. An obvious use case would be programmable frequency filters, when perhaps a more complex frequency response is desired.

Overall, our work offers exciting new insights into the inner workings of Neural Processes and into the learning of representations of function spaces. Many applications of deep learning are concerned with representation learning in some way, and we hope that our findings inspire further research and forge a better understanding of the methods used in the field. Our work also opens up a number of exciting questions for future work. We only look at function spaces with scalar domain, and while we expect that our findings translate to higher dimensions, the same should be validated empirically. Seeing that variational Neural Processes can in principle learn frequency decompositions, it would be interesting to investigate how we can further incentivize this behaviour in them. Likewise, it should be possible to encourage orthogonality between the individual representation dimensions, so that frequencies are more cleanly separated. Further theoretical exploration of the conditions, besides frequency content, that allow function spaces to be represented could also be worthwhile. Finally, it is not immediately obvious how our findings translate to scenarios that disallow a classical definition of frequency, for example when the observations are entire images as in Eslami et al. (2018).

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

# A    APPENDIX

## A.1    OPTIMIZATION & IMPLEMENTATION

To train Neural Processes, we represent individual examples $f \in \mathcal{F}$ as sets of randomly sampled evaluations $(\mathbf{x}, f(\mathbf{x}) = \mathbf{y})$, which we partition into *context set* $(\mathbf{x_c}, \mathbf{y_c})$ and *target set* $(\mathbf{x_t}, \mathbf{y_t})$. We further have encoder $E$ and decoder $G$ of a Neural Process implemented as neural networks, for which we summarize the parameters in $\theta$. In our implementation, both are multilayer percep-trons (MLP), meaning simple fully connected networks. Our goal is then to find the optimal set of parameters $\theta^*$ that maximizes the likelihood of $\mathbf{y_t}$, given $\mathbf{x_c}, \mathbf{y_c}$ and $\mathbf{x_t}$, over all $f$:

$$\theta^* = \arg\max_\theta \sum_{f\in\mathcal{F}} \log p_\theta(\mathbf{y_t}|\mathbf{x_t}, \mathbf{x_c}, \mathbf{y_c}) \tag{3}$$

where $p_\theta$ is a placeholder for some parametrized likelihood function. We introduce the logarithm because we assume the likelihood factorizes across individual $f$, turning the expression into a sum. So what would this optimization look like in practice? For example, we could minimize the mean squared error between $\mathbf{y_t}$ and the predictions $\hat{\mathbf{y_t}}$ from our network. This would implicitly assume a Gaussian likelihood with a fixed variance. However, we would like our model to predict a variance, so that it can indicate how uncertain it is about a prediction, and because Le et al. (2018) found that this results in overall better performance. We achieve this by implementing $G$ as a network that predicts both the mean and the variance of a diagonal Gaussian distribution, and Eq. (3) becomes:

$$\theta^* = \arg\max_\theta \sum_{f\in\mathcal{F}} \sum_t \log \mathcal{N}(y_t; G_\theta^\mu(Z, x_t), G_\theta^\sigma(Z, x_t)) \tag{4}$$

In deterministic Neural Processes (CNP), we can directly optimize this with maximum likelihood training. In variational Neural Processes (NP), $Z$ is also parametrized by a Gaussian, meaning just like $G$, $E$ predicts mean and variance of a Gaussian with $D_r$ dimensions. In this case, we need to rewrite the summands of Eq. (3):

$$\log p_\theta(\mathbf{y_t}|\mathbf{x_t}, \mathbf{x_c}, \mathbf{y_c}) = \log \mathbb{E}_{z\sim p(Z|\mathbf{x_c}, \mathbf{y_c})} p_\theta(\mathbf{y_t}|\mathbf{x_t}, z) \tag{5}$$

Here, $p(Z|\mathbf{x_c}, \mathbf{y_c})$ is *not* the distribution predicted by our encoder, but some *true distribution* we don't have access to. The idea of *variational inference* (see for example Bishop (2006) for an introduction) is to approximate this $p$ by some other distribution $q_\theta$ and then to optimize $p_\theta$ and $q_\theta$ simultaneously. $q_\theta$ is what our encoder $E$ predicts, just like $p_\theta$ is what our decoder $G$ predicts. Continuing from Eq. (5):

$$\text{LHS} = \log \mathop{\mathbb{E}}_{z \sim q_\theta(Z|\mathbf{x_t},\mathbf{y_t})} p_\theta(\mathbf{y_t}|\mathbf{x_t},z) \frac{p(z|\mathbf{x_c},\mathbf{y_c})}{q_\theta(z|\mathbf{x_t},\mathbf{y_t})} \tag{6}$$

$$\geq \mathop{\mathbb{E}}_{z \sim q_\theta(Z|\mathbf{x_t},\mathbf{y_t})} \log \left( p_\theta(\mathbf{y_t}|\mathbf{x_t},z) \frac{p(z|\mathbf{x_c},\mathbf{y_c})}{q_\theta(z|\mathbf{x_t},\mathbf{y_t})} \right) \tag{7}$$

$$\approx \mathop{\mathbb{E}}_{z \sim q_\theta(Z|\mathbf{x_t},\mathbf{y_t})} \log \left( p_\theta(\mathbf{y_t}|\mathbf{x_t},z) \frac{q_\theta(z|\mathbf{x_c},\mathbf{y_c})}{q_\theta(z|\mathbf{x_t},\mathbf{y_t})} \right) \tag{8}$$

$$= \mathop{\mathbb{E}}_{z \sim q_\theta(Z|\mathbf{x_t},\mathbf{y_t})} \log p_\theta(\mathbf{y_t}|\mathbf{x_t},z)$$
$$- D_{KL}(q_\theta(z|\mathbf{x_t},\mathbf{y_t})||q_\theta(z|\mathbf{x_c},\mathbf{y_c})) \tag{9}$$

where LHS refers to the left hand side of Eq. (5). In the first line, we have switched the underlying distribution from the true *prior*—meaning conditioned on the context—to an approximate *posterior*—meaning conditioned on both context and target, but for notational simplicity we only write out the target set. The second line follows from Jensen's inequality while in the third line we have replaced the true prior with the approximate prior. Finally, we have rewritten the right hand side using the Kullback-Leibler (KL) divergence, a measure of distance between two distributions. Because we predict Gaussian distributions, the KL divergence has a closed-form expression. Otherwise it would be impractical to use it in an optimization context. The last line is often called the *evidence lower bound* (ELBO) in variational inference.

Let us put the above into more practical terms. When presented with an example consisting of context and target sets, we first use the encoder network $E$ to encode each context tuple separately. The encoder is a MLP with two input channels (for $X$ and $Y$), 6 hidden layers with 128 channels, and a final layer mapping to $D_r$ channels, i.e. to the representation. While all hidden layers have a fixed dimension of 128, we vary the representation dimension $D_r$ for our experiments (but never make it larger than 128). For the variational case, the final layer maps to $2D_r$ channels, half for the mean and half for the variance of the predicted Gaussian (in practice, we predict the log-variance to allow negative values). The individual representations are then averaged, and in the variational case we call this the prior ($q_\theta(z|\mathbf{x_c},\mathbf{y_c})$ in Eq. (9)). For the posterior, we also encode the target pairs and then average over all individual representations, including the context. During training forward passes, we sample once from the posterior and use this sample as the representation for the decoder. Ideally, we should sample many times to integrate the expectation in Eq. (9), but for stochastic mini-batch training it was found empirically that a single sample suffices (Jimenez Rezende et al., 2014; Kingma & Welling, 2014). The decoder predicts a Gaussian from the representation and an input $x_t$. It is implemented symmetrically to the encoder, meaning it's a MLP with $D_r + 1$ input channels, 6 hidden layers with 128 channels, and two output channels for mean and (log-)variance. We use $\tanh$-activations as well. As a loss we directly use the negative log-likelihood, meaning we evaluate the likelihood of a reference point $y_t$ under a Gaussian parametrized by the predicted mean and variance. Finally, we average over all predicted points, which are the target points as well as the context points. We use the Adam optimizer Kingma & Ba (2015) with an initial learning rate of 0.001, repeatedly decaying it with a factor of 0.995 after 1000 batches. We train with a batch size of 256 for a total of 600 000 batches.

## A.2 EXPERIMENT DETAILS

We conduct our experiments on two kinds of function spaces. The first is defined by a Gaussian Processes prior using an EQ kernel given by:

$$k(x_1, x_2) = \exp \left( \frac{||x_1 - x_2||_2^2}{2l} \right) \tag{10}$$

where $l$ is a lengthscale parameter. This example was also used in the original works (Garnelo et al., 2018a;b). The second are random Fourier series, defined by:

$$f(x) = a_0 + \sum_{k=1}^{K} a_k \cos\left(kx - \phi_k\right) \ , \quad K = 19 \tag{11}$$

where we sample $\phi_k$ and $a_k$ (including $a_0$) randomly from the interval $[-1, 1]$. Note that $k$ is an angular frequency, while results are presented for regular frequencies.

To construct training examples, we sample $N$ context inputs and $M$ target input values uniformly from the range $[-3, 3]$. $N$ is a random integer from the range $[3, 100)$, while $M$ is a random integer from $[N, 100)$. This sampling strategy was adopted from the original works and Le et al. (2018). y-values are generated by evaluating the above functions (or drawing from the distribution in the case of the GP) on the random input values. The models are trained by letting them predict/reconstruct both context and target points, conditioned only on the context. At test time, we are only interested in reconstructions, meaning target points and context points are identical, and we work with 200 equally spaced input values across the full range.

In the band filter experiment, we train models on Fourier series with some frequencies intentionally left out of the training data. When we train a model on data where some frequency components are blocked, the distribution of y-values a model sees during training becomes narrower. As a result, passing functions from the reference distribution (where no components are blocked) through a band-filter CNP will suppress the desired frequencies, but will also *amplify* non-blocked frequencies. To counteract this, we have to multiply the y-values of the reference data, which are approximately normally distributed, by $\sigma_{\text{band}}/\sigma_{\text{ref}}$, i.e. the ratio of standard deviations of the relative y-distributions.

## A.3 Additional Visualizations

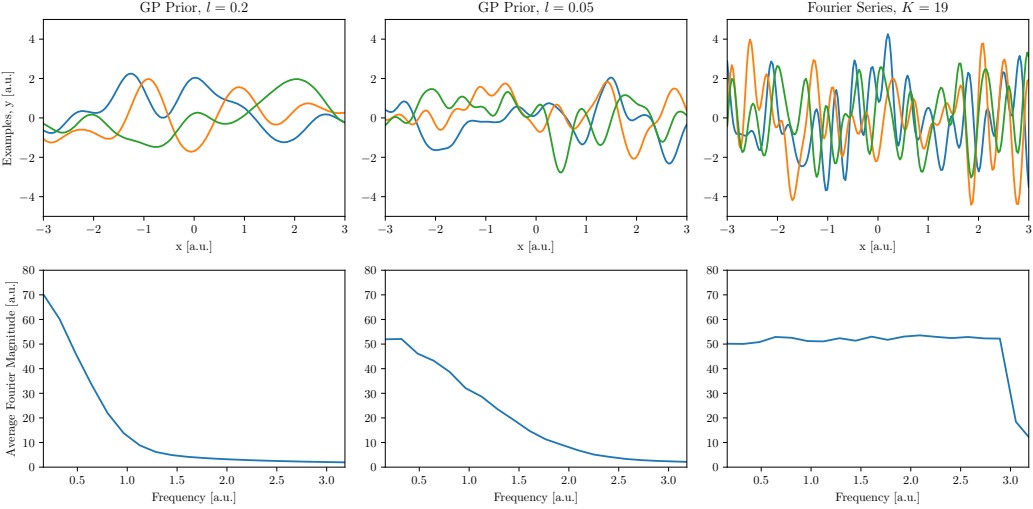

Figure A.1: Examples from the function spaces employed in this work (top row) and the corresponding average Fourier magnitudes (bottom row), averaged over 1000 samples. The Gaussian Process kernel and the expression for the Fourier series are given in Eq. (10) and Eq. (11).

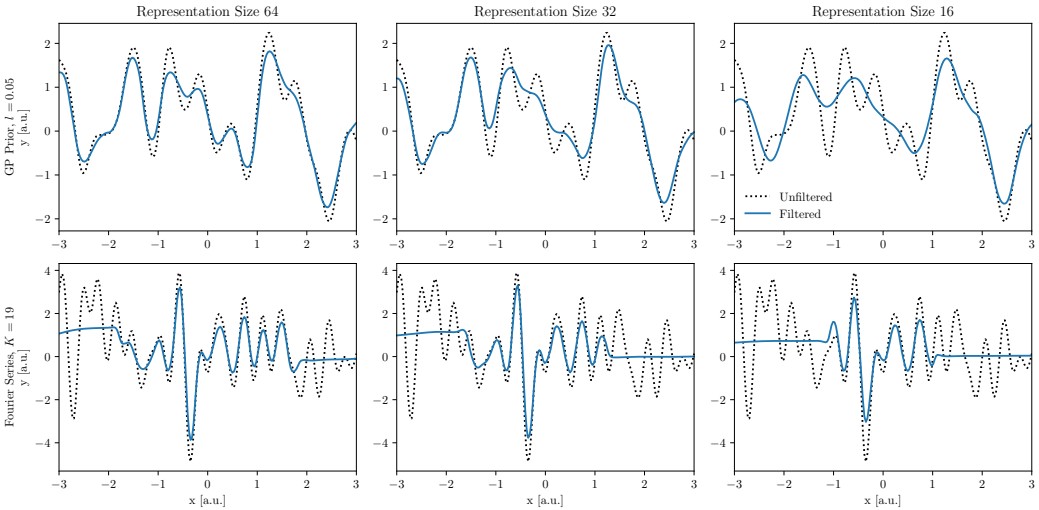

Figure A.2: Example reconstructions from a variational Neural Process (NP) with varying representation sizes. For the Gaussian Process example, the NP essentially behaves like the CNP as seen in Fig. 1, but with stronger dampening of the higher frequencies. Like the CNP, it also limits the effective interval for the Fourier series example, but again with stronger dampening of high frequencies.

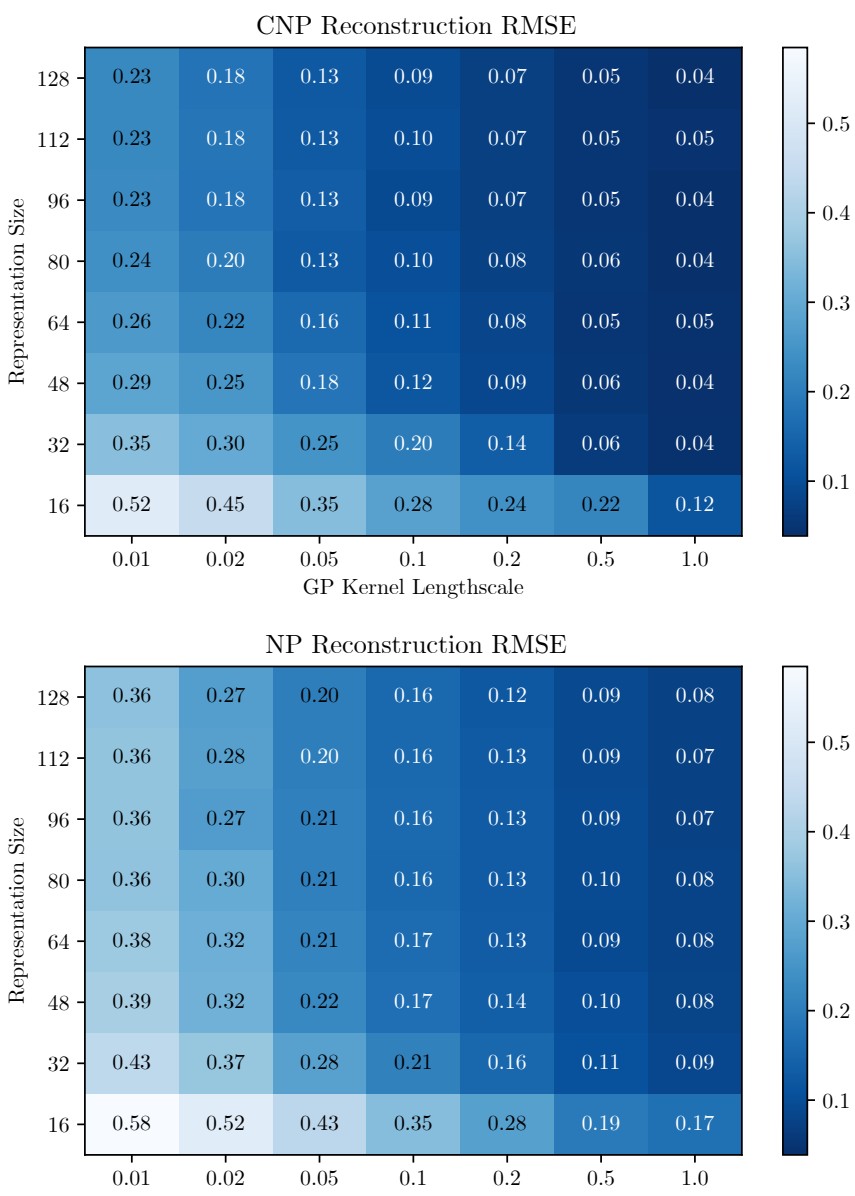

Figure A.3: Reconstruction performance (RMSE) of CNPs (top) and NPs (bottom) on data generated from a GP prior with an EQ kernel. We vary the kernel lengthscale—a smaller lengthscale means higher frequency content—and the representation size in the Neural Processes. Evidently, a larger representation allows the models to better reconstruct data with smaller lengthscale. Overall, the RMSE in CNPs is a little lower than in NPs.

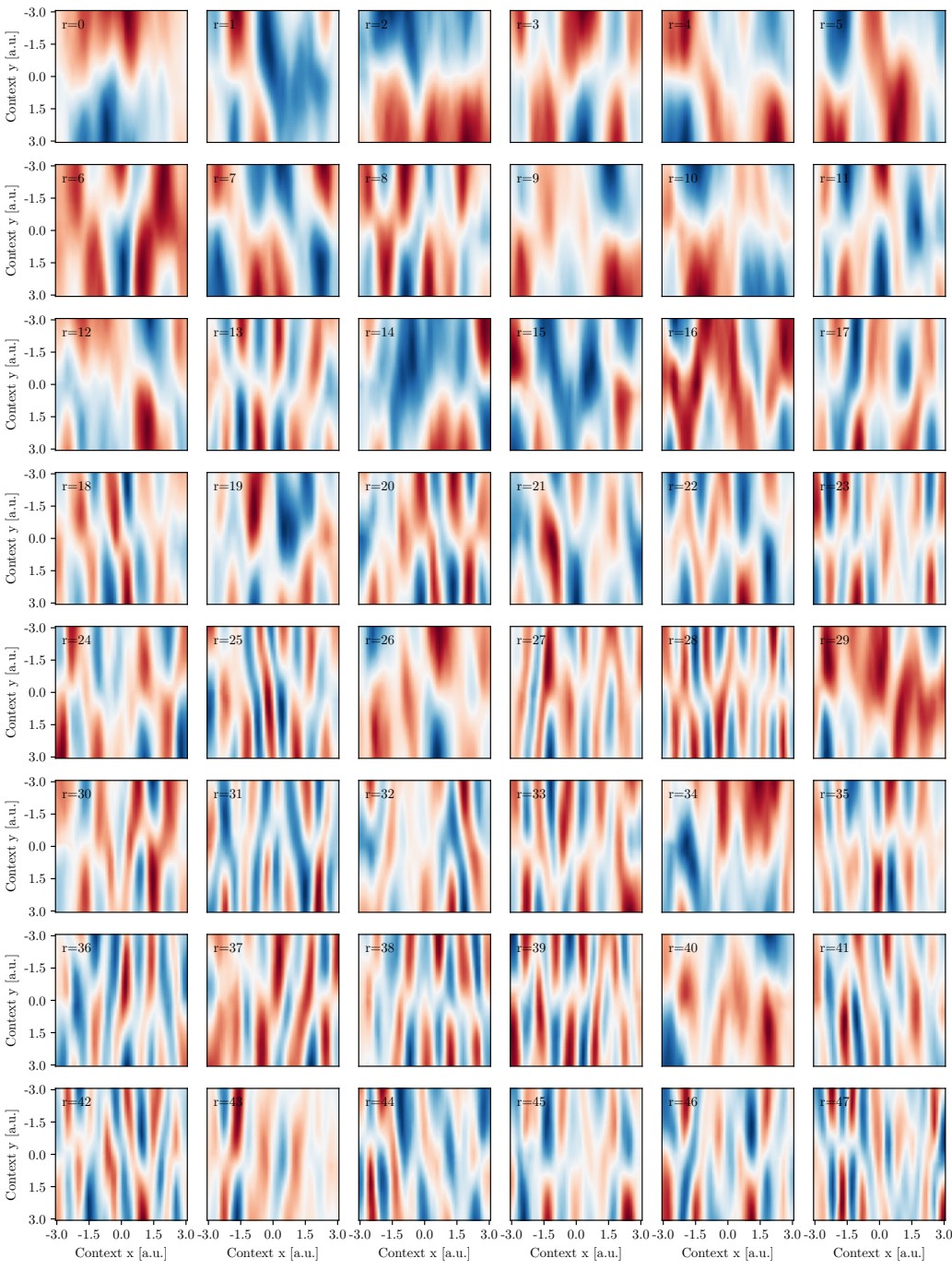

Figure A.4: Influence of the context on the learned representations in a deterministic Neural Process (CNP) with $D_r = 128$ and data coming from a Gaussian Process prior with an EQ kernel with $l = 0.2$. The description of the kernel is given in Eq. (10). $x$ refers to the input space (i.e. the domain), $y$ to the output space (i.e. the co-domain). These are the first 48 representations ordered by their average magnitude (left-to-right, top-to-bottom). Note that each panel is normalized separately, so color values are not comparable. While the representations are more or less anti-symmetric across $y = 0$, they exhibit an oscillating pattern along the x-axis. The frequency of those oscillations varies for the individual representation channels, leading us to conclude that CNPs perform a frequency decomposition of the input space.

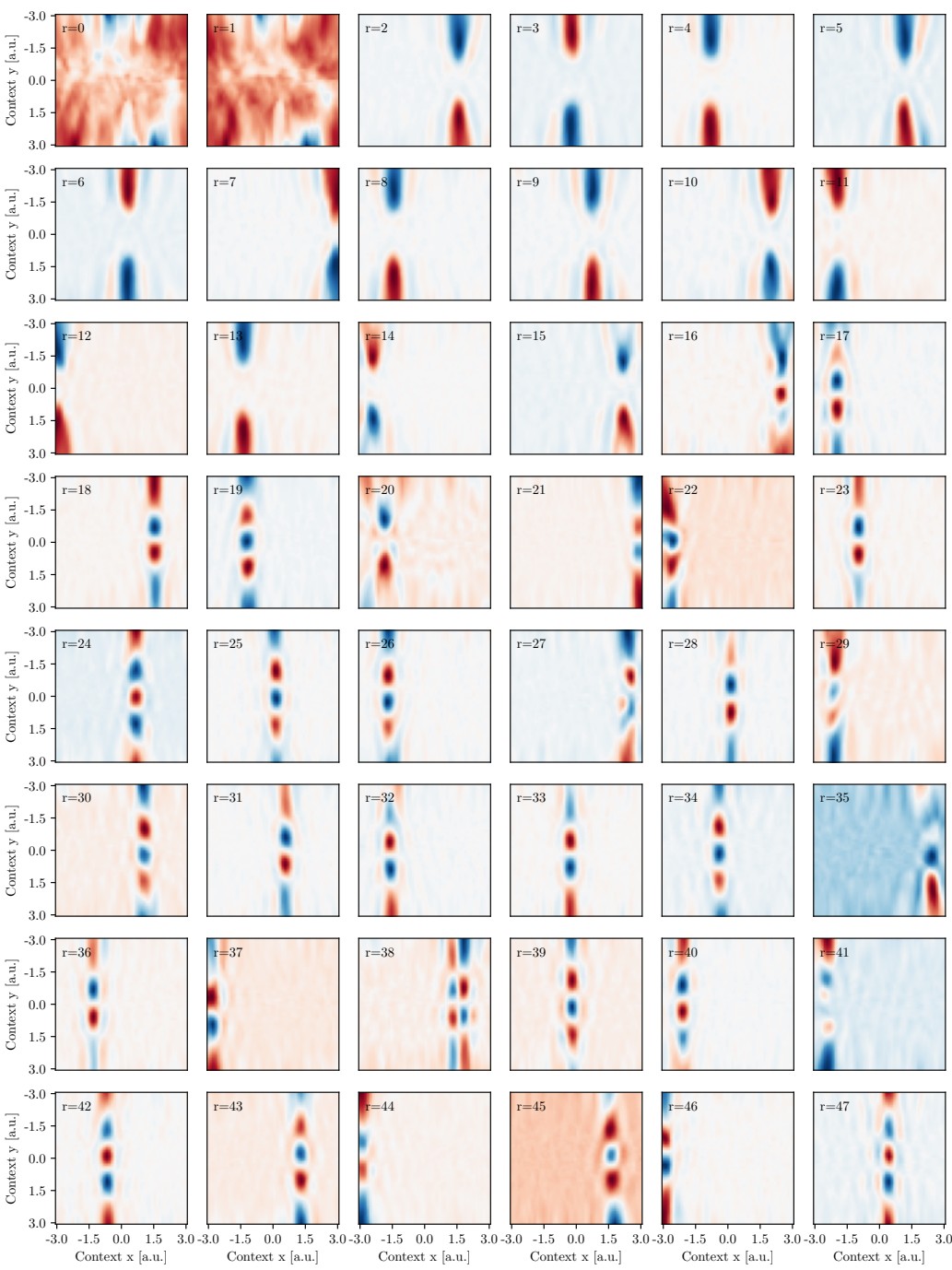

Figure A.5: Influence of the context on the learned representations in a variational Neural Process (NP) with $D_r = 128$ and data coming from a Gaussian Process prior with an EQ kernel with $l = 0.2$. The description of the kernel is given in Eq. (10). $x$ refers to the input space (i.e. the domain), $y$ to the output space (i.e. the co-domain). These are the first 48 representations ordered by their average magnitude (left-to-right, top-to-bottom). Note that each panel is normalized separately, so color values are not comparable. Unlike the representations learned in a CNP seen in Fig. A.4, the NP seems to partition the input space into almost discrete regions.

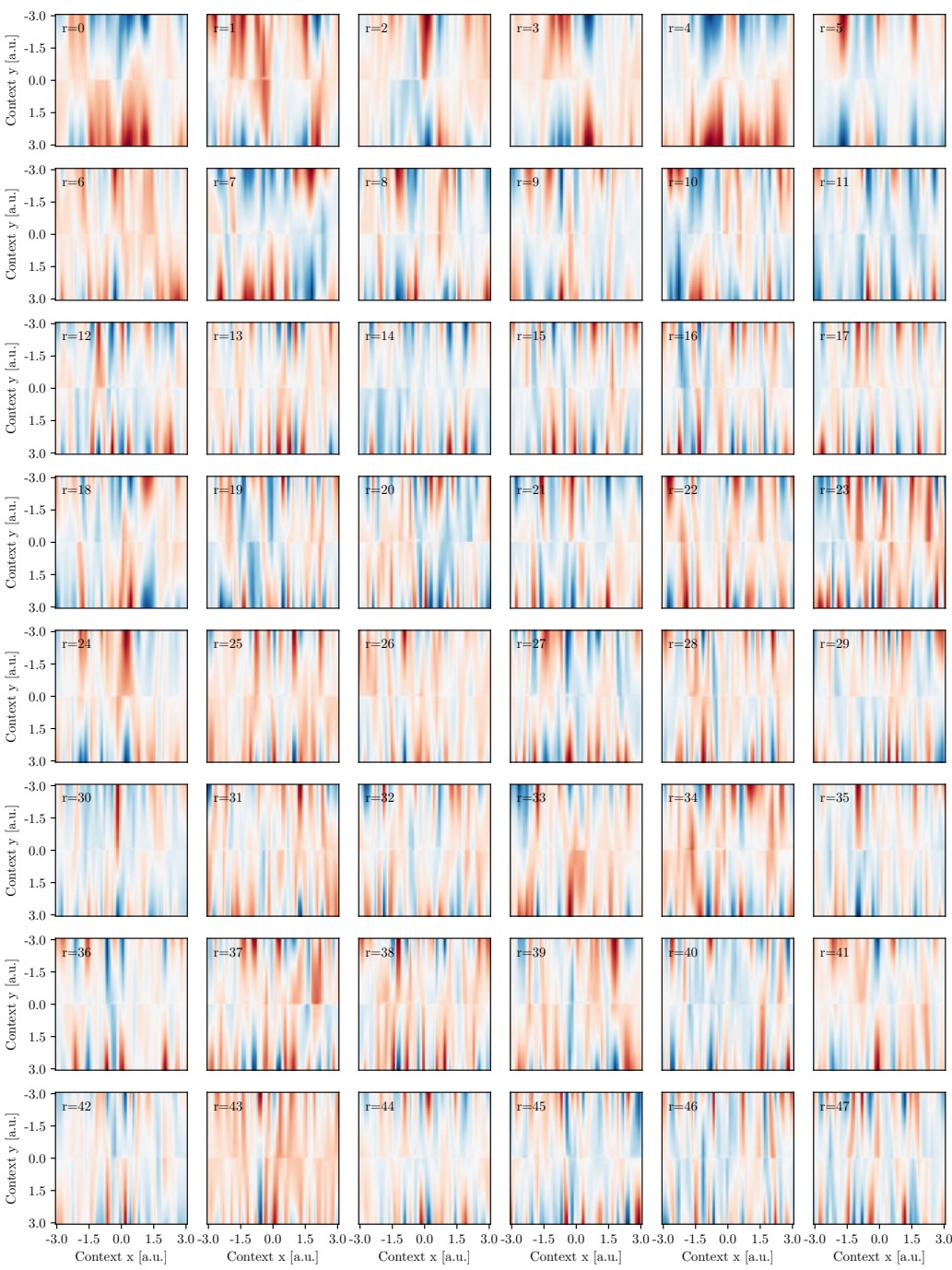

Figure A.6: Influence of the context on the learned representations in a deterministic Neural Process (CNP) with $D_r = 128$ and data coming from a random Fourier series with $K = 19$ as described by Eq. (11). $x$ refers to the input space (i.e. the domain), $y$ to the output space (i.e. the co-domain). These are the first 48 representations ordered by their average magnitude (left-to-right, top-to-bottom). Note that each panel is normalized separately, so color values are not comparable. Similar to the GP example in Fig. A.4, the CNP exhibits an oscillating pattern along the x-axis. The individual frequencies are not cleanly separated by the representation channels.

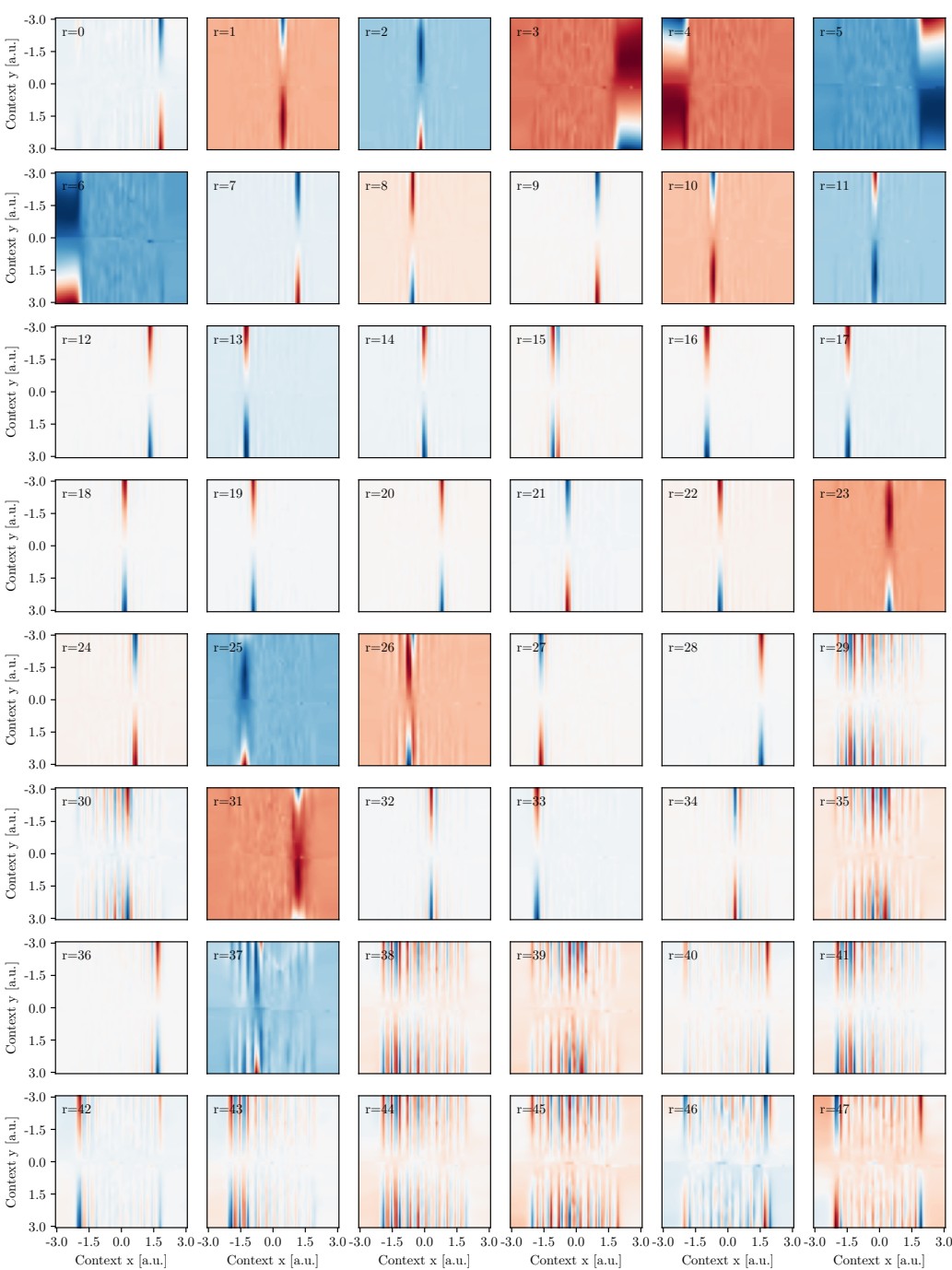

Figure A.7: Influence of the context on the learned representations in a variational Neural Process (NP) with $D_r = 128$ and data coming from a random Fourier series with $K = 19$ as described by Eq. (11). $x$ refers to the input space (i.e. the domain), $y$ to the output space (i.e. the co-domain). These are the first 48 representations ordered by their average magnitude (left-to-right, top-to-bottom). Note that each panel is normalized separately, so color values are not comparable. Similar to the GP example in Fig. A.5, the NP partitions the input space into virtually discrete regions, exhibiting even narrow and more sharply separated regions. Representations with a higher index also show oscillating behaviour similar to a CNP, indicating that it's in principle possible for NPs to learn frequency decomposition.

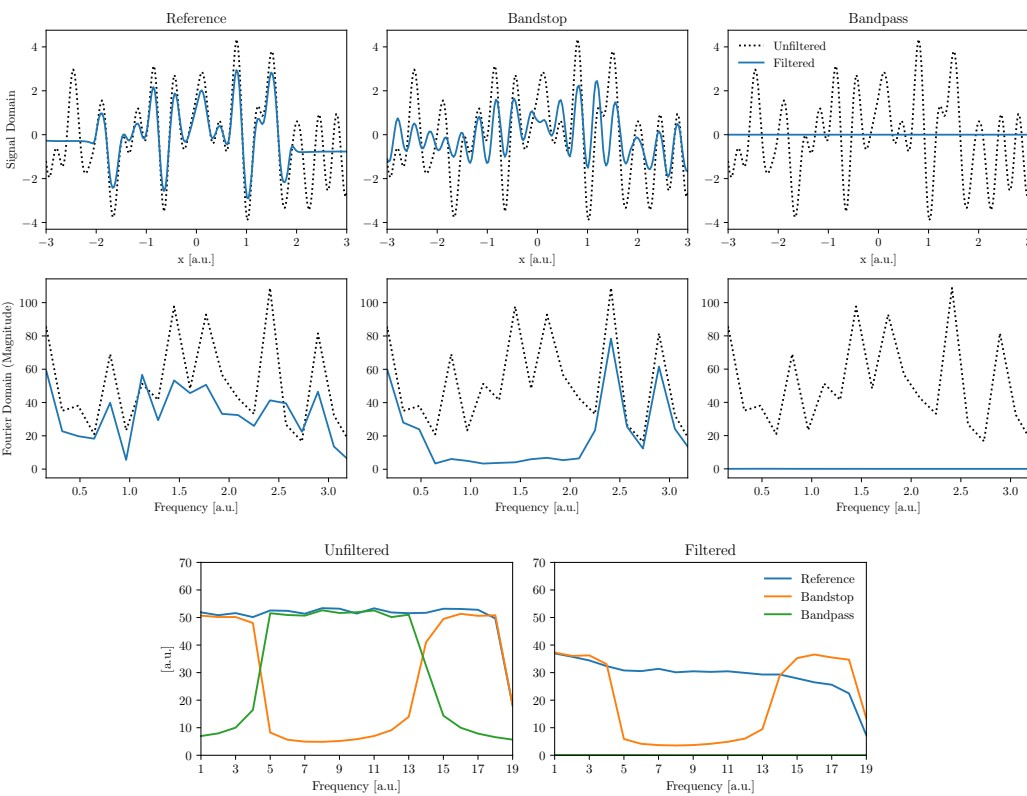

Figure A.8: Variational Neural Process (NP) trained to be a band-pass and a band-stop filter. We train three NP models on Fourier series, where we only show certain frequencies during training for the models that should serve as frequency filters. The bottom row shows the distribution of Fourier components in the training data (left) and after applying each model to the *reference* data (right). While the band-stop NP appears to work quite similar to the CNP seen in Fig. 3, we were not able to successfully train a band-pass NP.

