# OpenReview forum: "Frequency Decomposition in Neural Processes"
_ICLR.cc/2021/Conference — Reject_

### Official Review · AnonReviewer4 · 2020-10-27
**Recommendation to reject**

**Rating:** 3
**Confidence:** 4

**Review:**

This paper presents an analysis on the neural processes in the signal processing point of view and gives a bound on the highest frequency of the function that a neural process can represent.

I recommend to reject this manuscript. My comments are below.

The key point of this work is Theorem 3.1. However the theorem itself is just a direct outcome of the Nyquist–Shannon sampling theorem, and it is generally true to not only neural processes but also to all the other approaches. Meanwhile, the authors did not talk about the relationship quantitatively between the representability and the error tolerance in Definition 3.1. In addition, the analysis is limited to only scalar-valued function on a 1D interval. The writing could also be improved.

Concerns:
- The definition of neural processes in the background section is confusing. Despite the way of defining a map, P is a mathematical object defined by a set of tuples and a map,  meaning that the neural processes are also defined by data. In the original paper, the neural processes were however defined as random functions.

- In the background section, the words say 'some sources define ...'. Could the authors give the sources?

- In Def 3.1, what do the authors mean by 'discrete measurements'?

- In the experiment section, do the authors mean sampling from a Gaussian process by saying GP prior? I don't see a GP plays the role of prior in terms of Bayesian inference.

- The examples given in the experiment section lack quantitative results. It is better for evaluating the reconstruction by showing the posterior or predictive distribution instead of single reconstructions.

- In Sec. 4.2. how did the authors sample regular grid on the 2D plane as y is determined by x.

- Eq.11 is defined in the appendix. Better to use separate numbering.

---

> ### Author Response · Authors · 2020-11-16
> **Response to Review**
>
> Hi! Thanks for taking the time to review our paper!
>
> It is true that the Nyquist-Shannon theorem is not specific to NPs, but we don't see how that takes away from our contribution? The derivation of our theorem might not seem complicated in hindsight, but the key was to recognize that finding in Wagstaff et al., which concerns arbitrary set-valued functions, can be combined with Nyquist-Shannon in scenarios where the sets contain evaluations of continuous functions, as is the case here. We did not evaluate the error tolerance quantitatively because a) it's usually something that would depend on a particular use case; b) it's not really relevant to the main finding that NPs automatically find a representation in function space and c) the results would essentially be the same as those presented in the appendix, where we show the reconstruction error as a function of representation size and GP kernel lengthscale. You point out correctly that we only look at scalar functions, and we agree that behaviour for higher dimensional observations is an interesting avenue for future work (we do acknowledge this in the discussion). However, the analyses we perform here would have been very hard to interpret for higher-dimensional problems, which is why we decided to stay in 1D.
>
> Allow us to answer your other points in an itemized fashion as well:
> * The derivation of NPs in the original paper was probably made in a way to be close to GPs, but in the end it is stated that NPs learn a distribution over random functions (p.2 second to last paragraph), which is essentially the same as what we say. The NP is obviously defined by data, as is virtually every DL model, but the trained NP model is in fact a map from C, X to Y, as we state in our paper.
> * We will add exemplary sources in the final version / next iteration, thanks for the suggestion
> * Discrete measurements just means that we have individual points of a signal/function that is assumed to be continuous. Does that answer your question?
> * Yes, we mean sampling from a Gaussian Process, but we say prior to make it clear that it is not the posterior, which would already be conditioned on some points. The GP is just used as one source of random functions, it doesn't have any other meaning (Bayesian or other) in our context.
> * That's a good point, we will address this in the next version: the predictive distribution is almost always narrow enough to no be visible in the plots, but we should have stated that of course.
> * Yes, y is a function of x, but think of all the functions y(x) of a given distribution/function space occupying a certain area of the (x,y) space. We now sample this space with both x and y going from -3 to 3 in 50 steps (think of np.linspace) and then just encode all those points to create a visualization of r_i(x, y) to understand how that space is now represented in the NP.
> * Thanks for the suggestion
>
> We hope we were able to answer your questions!
>
> Kind regards, the authors

---

> > ### Comment · AnonReviewer4 · 2020-11-16
> > **Not to flip recommendation**
> >
> > Thanks for the authors' response.
> >
> > Though the NS bound is true, the novelty is still unclear. Regarding the relationship between the representability and the error tolerance in Definition 3.1, my concern was, since the NS bound has nothing to do with the error tolerance, it would be a significant contribution if you can relate them in the context of NPs.
> >
> > Does the comma in "a map from C, X to Y" indicates the Cartesian product?

---

> > > ### Author Response · Authors · 2020-11-17
> > > **Thanks for the Clarification**
> > >
> > > So essentially you'd like to see the tightness of the bound evaluated, as a function of error tolerance? That's certainly a good point, thank you. We did try to evaluate the tightness of the bound in general, but haven't yet come up with a way that's as rigorous as we'd like it to be.
> > >
> > > Yes, the comma is meant to be a cartesian product.
> > >
> > > Thanks again!

---

### Official Review · AnonReviewer2 · 2020-10-28
**Interesting approach, but insufficiently supported claims**

**Rating:** 4
**Confidence:** 3

**Review:**

This paper addresses an interesting and timely problem, which is to understand how Neural Processes work to learn a representation of a function space. Offering a closer investigation into a recently introduced framework, this work will likely be of interest to the ICLR community. The work focuses on the 1-dimensional case and tries to analyze the simplest case in a rigorous way, which I think is a good approach in general.

However, I have some concerns about the main claims of this paper, as listed below:

- One of the main findings of the paper is an observation that Neural Processes perform a "frequency decomposition". However, I think this is an insufficiently supported, and even misleading, over-statement. Indeed, Figure 2 shows that there are different modes dominated by varying characteristic frequencies, where a higher-rank mode shows a more slowly varying feature; but there is no further evidence that the decomposition is actually based on the frequency of the signal. One would get a similar result by simply doing a Principal Component Analysis too. When you say "frequency decomposition" it carries a clear mathematical meaning, and it is a much stronger statement than what the paper reports empirically.
    - That said, I agree that the empirical observations are interesting. Perhaps the observations in the paper's experiments may be better described in a frame of global mode decomposition (CNP) vs. local feature detection (NP)?

- I also think that the claim about the theoretical upper bound on the frequency is overstated, the way it is stated currently. The validity of the statement (Theorem 3.1) really depends on the assumption of uniform sampling, which is mentioned as a note after Theorem 3.1. Of course, I fully agree that it is an important starting step to get rigorous results in simplified conditions. But those conditions should be mentioned as part of the statement, especially when it is highly likely that the conditions are not met in the use case (there is no reason to expect that the x values in the context set is close to uniform). For example, it is possible to encode functions with a localized feature whose (local) frequency is higher than your derived bound, by using more samples around that high-frequency feature.

This paper will get views, partly because it is actually asking an interesting question, and partly because of the boldness and attractiveness of the claims made. How exciting is it to discover a naturally emerging Fourier transform? Except... that's not exactly what one can say just yet (I think). I believe the authors should either support the paper's claims by further work, or tone down their overall framing — major changes either way. While I think this work is headed to a promising direction, given the concerns described above, I recommend a rejection at this time.

**UPDATE:** I appreciate the authors' responses and the engaged discussion. However, I still think that the claims of the paper are not sufficiently supported by the presented results, and maintain my original rating.

---

> ### Author Response · Authors · 2020-11-16
> **Response to Review**
>
> Dear R2, thank you four your detailed input,
>
> regarding your first point, we agree to the extent that the visualization of the representations can only be considered qualitative evidence for the hypothesis that a frequency decomposition occurs. That is why we included the band filter experiment, which should only be possible if the representations are indeed learned in frequency space. Do you have suggestions on how to show more directly that this is a frequency decomposition? And could you elaborate on your comment that a similar result would be achieved with a PCA? We don't see how that's the case.
>
> On the concern regarding the equidistant sampling requirement in the Nyquist-Shannon theorem, that's not actually a hard requirement, signals can also be reconstructed from random sampling points if the average sampling rate meets the Shannon limit. Because we sample x from a uniform distribution, we can thus use the same derivation as if they were equidistantly sampled. We do agree that we should have discussed this in more detail! Your idea to test the behaviour with non-uniform sampling is very interesting and we will look into it.
>
> We'd also like to thank you for your encouraging words on the direction of our work, they do mean a lot :)
>
> Kind regards, the authors

---

> > ### Comment · AnonReviewer2 · 2020-11-17
> > **Still don't think that this is a frequency decomposition**
> >
> > Dear authors, thank you for the response. To answer your comments first:
> >
> > > … the band filter experiment, which should only be possible if the representations are indeed learned in frequency space.
> >
> > I don't quite agree… a model could achieve the same result by simply learning a dictionary whose elements happen to consist of functions of different timescales.
> >
> > > could you elaborate on your comment that a similar result would be achieved with a PCA?
> >
> > If one performs a PCA on some smooth (but mixed-frequency) synthetic timeseries, for example generated from a gaussian process, I wouldn't be surprised to find that the PC1 resembles a "low-frequency" mode, and the following PCs increasingly "higher-frequency".
> >
> > I am not changing the original recommendation; I still think that this result is not really a frequency decomposition. There should be a better, and more accurate, way of describing the findings. The CNP result finds modes with different characteristic timescales, but they are not identified with any specific frequencies, nor have any reason to be so. Maybe one could say that these modes are more strongly _localized_ in frequency than in time.

---

> > > ### Author Response · Authors · 2020-11-17
> > > **Thanks for the Clarification**
> > >
> > > Dear R2, thanks for elaborating,
> > >
> > > > a model could achieve the same result by simply learning a dictionary whose elements happen to consist of functions of different timescales.
> > >
> > > Ok, but if a signal is now (re-)constructed as a combination of those dictionary elements, which are functions of different timescales (=functions with different frequency), that's exactly what we're saying... Is the point here that we can't be sure that the signal is actually "decomposed" in any way, but could also be memorized?
> > >
> > > Thanks for elaborating more on the PCA point, we get what you mean! It could be that a PCA would behave similarly, but we can't be certain until we try, so we'll look into it. If it does, however, we still don't see how that would invalidate our findings or make them overly obvious.
> > >
> > > Thanks again!

---

> > > > ### Comment · AnonReviewer2 · 2020-11-19
> > > > **Response to authors**
> > > >
> > > > Dear Authors,
> > > >
> > > > I have absolutely no problem calling this dictionary example (and also your finding in the paper) a decomposition. What I feel uncomfortable about is calling it a _frequency_ decomposition. When an operation is called a "frequency decomposition", I think it is strongly suggestive of a _mechanism_ that is based on the notion of frequency, which is not what you are presenting. I feel that the more appropriate way to talk about these findings would be _descriptive_: you discovered an interesting decomposition (I agree), and you could describe that the modes have different characteristic frequencies (I see that).
> > > >
> > > > To be fair, though, I came to realize that this may be a subjective interpretation of the language, possibly related to the fact that I was trained as a physicist.

---

> > > > > ### Author Response · Authors · 2020-11-19
> > > > > **Thanks for the discussion**
> > > > >
> > > > > Dear R2,
> > > > >
> > > > > thanks for taking the time to discuss (even though our paper will certainly be rejected)!
> > > > >
> > > > > What we were trying to show is that the NP _becomes_ this mechanism, without any external incentive, and if we understood you correctly, we're not quite managing to do that, but if we did, it would in fact be a very interesting contribution?
> > > > > In your definition, would a frequency decomposition only be one that actually separates _individual_ frequencies?  Is it one that uses sin/cos, i.e. essentially a Fourier transform? We're currently trying to enforce orthogonality in the representations, if we can then show that the representation is approximately a Fourier basis, would you say that our claim is validated?
> > > > > Or is the problem rather the fact that we only present evidence that _suggests_ a frequency decomposition is learned, but doesn't _prove_ it?
> > > > > * Frequency bound -> Suggests a notion of frequency, but can probably be observed in other kinds of representations.
> > > > > * Visualizations of Representations -> Looks like frequency decomposition (depending on definition), but certainly only a qualitative result.
> > > > > * Band filter -> In our opinion very strong evidence that learned representations reside in frequency space, but probably not impossible to get this behaviour in other ways.
> > > > >
> > > > > Sorry if we're a bit slow to grasp your point.

---

> > > > > > ### Comment · AnonReviewer2 · 2020-11-24
> > > > > > **Response to authors**
> > > > > >
> > > > > > Dear authors, I can't say much about the results that are not shown, but yes - I think what could be called a frequency decomposition would naturally look very much like a Fourier transform. If you show that that's what NP does, indeed it would be an interesting contribution. However, from what I understand, I feel that it is most likely not the case; frequencies may not be the essence of your findings. I may be wrong, but for the sake of completing the discussion let me try to make the point one last time. Suppose that your signals are local bumps in the time space, of fixed widths but varying center positions and heights. Would the decomposition (i) find a basis that includes single bumps at various positions, so that each signal is explained by ~1 of these, or (ii) a bunch of frequency modes where each signal is a superposition of _many_ modes? I would call only (ii) a frequency decomposition, but is that what NP does?

---

> > > > > > > ### Author Response · Authors · 2020-11-24
> > > > > > > **Response**
> > > > > > >
> > > > > > > Dear R2, thanks for the clarification. We still think that (ii) is what actually happens, hopefully we will find a way to demonstrate it more clearly :)

---

### Official Review · AnonReviewer1 · 2020-10-29
**Interesting paper on neural processes, unsure about applications and robustness of experiments**

**Rating:** 5
**Confidence:** 3

**Review:**

The paper tries to analyze the behavior of Neural Processes in the frequency domain and concludes that such Processes can only represent oscillations up to a certain frequency.

While drawing a parallel between Neural Processes and signal processes, I think that there is some weakness in the experiments of the paper. In particular, the authors only seem to consider the exponential quadratic kernel to generate examples which would mostly show examples of smooth functions as would sampling Fourier linear combinations.

I am also unsure how this paper could be helpful to our community in its present form as it sheds some light on the inner workings of Neural Processes but only in a very limited practical setting.

---

> ### Author Response · Authors · 2020-11-13
> **Response to Review**
>
> Hi! Thanks for taking the time to read and review our paper :)
>
> Regarding your comment on the types of functions we used, the choices were relatively straightforward: the GP samples were used in the original NP work, and the Fourier series gave us control about the frequencies in the signals. But we agree, more variety would have undoubtedly strengthened our paper. The implementation we use (fully connected + tanh activation) only allows for smooth functions, but it's certainly worth exploring other implementations as well!
>
> With respect to how our work can be helpful for the community, we believe it contributes significantly to the general understanding of Neural Processes. To the best of our knowledge, there is no other publication that tries to understand how function spaces are represented in these models. We do suggest one possible practical benefit of our discovery (learnable band filters), but our main contribution is still the finding that function spaces are represented via a frequency decomposition that emerges automatically, not so much the question how that translates to practical use of NPs. Is that what you mean by "limited practical setting"?
>
> Kind regards,
> the authors

---

> > ### Comment · AnonReviewer1 · 2020-11-18
> > **Response to authors**
> >
> > Dear authors,
> >
> > Thank you for your clarifications.
> >
> > In response to ` "is that what you mean by "limited practical setting"?", yes indeed.
> > I agree that your work is useful for the academic community it belongs to but at the end of the day, it would be interesting to also mention practical applications.

---

### Official Review · AnonReviewer3 · 2020-10-30
**Empirical studies confirm theoretical conclusions; however, what are the consequences?**

**Rating:** 6
**Confidence:** 2

**Review:**

The work examines properties of Neural Processes (NP). More precisely, of deterministic NPs and how they for finite-dimensional representations of infinite-dimensional function spaces. NP learn functions f that best represent/fit discrete sets of points in space. Based on signal theoretic aspects of discretisation, authors infer a maximum theoretical upper bond of frequencies of functions f that can be used to represent the points. The bond depends on the latent dimension/representation size and the finite interval spawn by the points. Simulations are computed to test the validity of the upper bond. Authors find that NPs behave like a Fourier Transform and decompose the spectrum of the signal. Since the representation during training learns to represent specific frequencies, NPs can be used as band pass/stop filter.

The paper is well written, and the basic approach is clearly outlined. The quality of the work and the evaluation are good and support the authors claims. However, it is not fully clear to which extend the claims translate to other data or generalise well. The finding that NPs interpret points in space as signals and implement a frequency decomposition like Fourier/Wavelet transforms seems reasonable. Not sure, however, if an application as filter is ecological in terms of computational complexity.

The paper provides a strong theoretical foundation of the method and authors support their claims by  empirical stimulation. Also, explainability and more importantly interpretability of how methods generate results is essential. So, the message the paper sends is relevant. However ,the relevance and significance of the findings, and the consequences thereof are not clear.

---

> ### Author Response · Authors · 2020-11-16
> **Response to Review**
>
> Dear R3, thank you for your review!
>
> regarding your question on the computational requirements w.r.t to a possible use of NPs as filters, that's certainly something that can be looked into further. Our goal was not to propose NPs as learnable band filters, we just thought that's a cool consequence of our main finding (that NPs automatically learn a frequency decomposition) and can be considered further evidence for it. Your second point---the significance and the consequences being unclear---seems to go in the same direction. And the truth is, we're not completely sure if there are any immediate use cases our contribution enables. It is more about the general understanding of how NPs represent function spaces, something that has never been looked into before. In our opinion, such advancements are as important as those that are focused more on practical utility.
>
> Thanks again and kind regards,
> the authors

---

> > ### Comment · AnonReviewer3 · 2020-11-24
> > **Thanks for clarifications**
> >
> > Thank you for the clarifications. After reading the authors replies and the comments of the other reviewers there are no more open questions. The approach is academically interesting; however, further analyses are needed to gain a clearer picture about the potential of the method.

---

### Decision · Program_Chairs · 2021-01-07
**Final Decision**

**Decision:**

Reject

**Comment:**

The paper analyses the behaviour of Neural Processes in the frequency domain and, in particular, how it suppresses high-frequency components of the input functions. While this is entirely intuitive, the paper adds some theoretical analysis via the Nyquist-Shannon theorem. But the analysis remains too generic and it is not clear it will be of broad interest to the community.